# Predicting Lung Function Using Biomarkers in Alpha-1 Antitrypsin Deficiency

**DOI:** 10.3390/biomedicines11072001

**Published:** 2023-07-15

**Authors:** Daniella A. Spittle, Alison Mansfield, Anita Pye, Alice M. Turner, Michael Newnham

**Affiliations:** Institute of Applied Health Research, University of Birmingham, Birmingham B15 2TT, UK

**Keywords:** alpha-1 antitrypsin deficiency, chronic obstructive pulmonary disease, lung function, biomarkers

## Abstract

Lung disease progression in alpha-1 antitrypsin deficiency (AATD) is heterogenous and manifests in different ways. Blood biomarkers are an attractive method of monitoring diseases as they are easy to obtain and repeatable. In non-AATD COPD, blood biomarker panels have predicted disease severity, progression, and mortality. We measured a panel of seven serum biomarkers in 200 AATD patients and compared levels between those with COPD and those without. We assessed whether biomarkers were associated with baseline lung function parameters (FEV1 and TLco) or absolute change in these parameters. In total, 111 patients with a severely deficient genotype of AATD (PiZZ) and COPD were included in the analyses. Pearson’s correlation coefficient was measured for biomarker correlations and models were compared using ANOVA. CRP and CCL18 were significantly higher in the serum of AATD COPD versus AATD with no COPD. Biomarkers were not predictive of cross-sectional lung function measurements, however, CC16 was significantly associated with an absolute change in TLco (*p* = 0.018). An addition of biomarkers to the predictive model for TLco added significant value over covariates alone (R^2^ 0.13 vs. 0.02, *p* = 0.028). Our findings suggest that CC16 is predictive of emphysema progression in AATD COPD. Proteomics data may reveal alternative candidate biomarkers and further work should include the use of longitudinal biomarker measurements.

## 1. Introduction

Alpha-1 antitrypsin deficiency (AATD) is an autosomal codominant inherited disorder characterised by low serum levels of alpha-1 antitrypsin, which causes a predisposition to lung and liver disease. It is classically associated with basal panacinar emphysema-predominant chronic obstructive pulmonary disease (COPD), which develops at an earlier age than usual (non-AATD) COPD, with obstructive lung disease typically being diagnosed between the age of 46–55 years [1]. AATD COPD is accelerated by smoking and environmental exposures, with tobacco smoke being the most important driver of lung disease in AATD subjects [2,3]. As well as having a diagnostic role, lung function tests reveal important physiological parameters relating to health status in COPD [4,5]. Forced expiratory volume in the first second (FEV1) is strongly predictive of all-cause mortality in COPD and its trajectory also carries important prognostic implications [6,7]. In addition, the transfer factor for carbon monoxide (TLco) has also been shown to predict health status and mortality in COPD [8,9]. In AATD-related COPD, identifying patients with poor lung function and those at risk of accelerated lung function decline is particularly relevant, as these patients should be optimised, which may include intravenous augmentation of plasma alpha-1 antitrypsin levels, although this treatment is not licensed in all countries [10,11]. Challenges remain in identifying patients at risk of accelerated lung function decline, as the decline is heterogenous with marked variation between patients [12]. Traditional risk factors are limited in their prediction, with no single clinical phenotype or functional assessment being able to fully capture the risk of poor long-term outcomes [13]; therefore, there is a need to identify additional prognostic factors. Serum biomarkers are an attractive option as they are easily obtainable and repeatable; furthermore, a panel of biomarkers may have a better predictive performance over individual biomarkers [14]. 

There have been promising advances in the identification of serum biomarkers as prognostic tools for lung function decline in usual COPD [14]. One of the most robust biomarkers described in COPD is the use of absolute blood eosinophil count (≥300 cells/µL) to endotype those at higher risk of exacerbations and those more likely to positively respond to inhaled corticosteroid treatment [15]. In a 10-year study of over 4000 subjects, club cell protein-16 (CC16) was identified as a promising biomarker of lung function and its progression, with the suggestion of its potential to identify rapid decliners for interventional studies [16]. Similarly, a prospective analysis of the Evaluation of COPD Longitudinally to Identify Predictive Surrogate Endpoints (ECLIPSE) cohort showed that surfactant protein-D (SP-D) was significantly associated with emphysema progression and mortality [17]. SP-D is considered an airway-specific protein, owing to its production by type II alveolar cells, and plays an important role in innate immune protection [18]. Another airway-associated protein of interest in COPD is chemokine ligand 18 (CCL18). CCL18 is largely expressed by alveolar macrophages in the lower airways and increased serum concentrations of the protein have been associated with emphysema, disease severity, and exacerbation severity [19,20,21]. 

Given that COPD is characterised by persistent airway inflammation, it is unsurprising that a number of inflammatory biomarkers are elevated in the serum of these patients. C-reactive protein (CRP), tumour necrosis factor alpha (TNF-alpha) and interleukin- (IL-) 6 and 8 are raised in COPD, with the latter two being associated with the progression of emphysema [22,23].

Despite the frequency of studies suggesting a role for individual biomarkers in the prediction of COPD outcomes, single biomarker measurements alone have limited predictive value [24]. An alternative approach, to add predictive value, is by measurement of multiple biomarkers in combination. Optimal combinations of five plasma biomarkers for the prediction of cross-sectional and longitudinal clinical outcomes were investigated in a large number of subjects in both the COPDGene and ECLIPSE cohorts [25]. Biomarker panels demonstrated additional predictive value of cross-sectional outcomes, particularly airflow limitation (FEV1) but showed little efficacy with longitudinal measurements [25]. 

In contrast, there is a paucity of evidence for AATD-related COPD. Limited post hoc longitudinal studies have identified serum proteins as potential biomarkers for AATD progression, such as metalloprotease-9, desmosine, CRP, adipocyte fatty-acid-binding protein, and tissue plasminogen activator [26,27,28]. However, only one of these studies evaluated lung function decline as a clinical outcome and it was limited in the number of biomarkers evaluated and the follow-up duration [26]. It is therefore vital that we draw on the more extensively investigated serum biomarker studies within usual COPD and investigate if they can predict lung function decline in AATD-associated COPD.

A panel of seven of the most robustly validated serum biomarkers for COPD were identified for inclusion, following a review of the literature. These are the systemic inflammatory markers CRP, IL6, IL8, and TNF-alpha, as well as the airway-specific markers SP-D, CCL18, and CC16. All of these biomarkers have been shown to have a significant association with FEV1 [21,22,25,29,30,31,32,33,34,35]. Most of these biomarkers have also been shown to predict FEV1 decline in longitudinal studies, with the exceptions of IL8, CCL18, and TNF-alpha [16,25,36,37,38]. IL8 and TNF-alpha have been shown to have a significant association with TLco [32]. This study of AATD patients aims to evaluate this panel of seven serum biomarkers, both individually and in combination, to predict baseline and longitudinal decline in FEV1 and TLco on lung function tests.

## 2. Methods

### 2.1. Study Design and Study Population

This retrospective cohort study sought to measure a panel of biomarkers in the serum of patients with AATD and their association with the lung function parameters FEV1 and TLco (Appendix A). Data from the original Birmingham registry were approved by the South Birmingham Research Ethics Committee (REC Ref: 3359a) and subsequently by the South-Central Oxford Research Ethics Committee (REC Ref: 18/SC/0541, IRAS Project ID: 233675). Patients attended a specialist AATD clinic between 2015 and 2018 and gave consent to be included in the Birmingham Alpha-1 registry at the Queen Elizabeth Hospital, Birmingham, UK, and this cohort was described in previous publications [39]. Post-bronchodilator spirometry or gas transfer factor was performed by respiratory physiologists according to quality-assured standards [40]. A total of 200 patients were used for biomarker analysis at a single time point and were selected based on the availability of lung function data (Appendix A). Patients underwent spirometry and gas transfer testing within one year of their serum collection. Longitudinal lung function data points were collected where available. Clinical information, including demographics and smoking status, was collected at all visits. 

### 2.2. Sample Collection and Biomarker Measurement

Blood samples were collected in BD Vacutainer^®^ serum tubes and centrifuged at 1000× *g* for 10 min. Serum was aliquoted and stored at −80 °C until required. Biomarkers were measured using commercially available enzyme-linked immunosorbent assay (ELISA) kits, according to manufacturer protocols. ELISA kits were selected with appropriate sensitivity, according to expected serum concentrations from the literature. All kits had been validated for use with human serum. CC16 was measured using Human Uteroglobin Quantikine ELISA Kit (R&D Systems, Minneapolis, MN, USA); CCL18 using Invitrogen PARC/CCL18 Human ELISA Kit (Fisher Scientific, Waltham, MA, USA); Invitrogen CRP Human ELISA Kit (Fisher Scientific); SP-D using Human Magnetic Luminex Assay (Bio-Techne Limited, Minneapolis, MN, USA); and IL6, IL8, and TNF-alpha using Human HS Cytokine A Premixed Mag Luminex Performance Assay (Bio-Techne Limited). Standard curves and interpolation of unknown concentrations were calculated using GraphPad Prism 8 version 9.0 (Boston, MS, USA). 

### 2.3. Statistical Analysis 

Statistical difference between biomarker concentrations in AATD with and without COPD was calculated using the Mann–Whitney U test. Biomarker correlations were measured with Pearson’s correlation coefficient (r). FEV1 and TL_CO_ absolute (litres and mmol/min/kPa, respectively) and percentage predicted values were predicted cross-sectionally at baseline (closest to the time of biomarker measurement), for the change in those absolute values (mLs/year and mmol/min/kPa/year, respectively), or percentage predicted values longitudinally. To calculate lung function change, patients with 3 or more lung function tests were included to enable slope to be calculated, which was then used in the subsequent multivariable models. Given that patients had values recorded at different time intervals, slope calculation standardised lung function decline. FEV1 and TL_CO_ (absolute and percentage predicted) were modelled using multi-variable linear regression adjusted for co-variables (age, sex, and smoking status) where appropriate. Any potential significant difference between models was compared using analysis of variance (ANOVA). Models were checked for normality of residuals, heteroskedasticity, and multicollinearity (variance inflation factor).

## 3. Results

### 3.1. Cohort Overview

Of 200 patients from this study cohort, 167 (83.5%) had ZZ-AATD and 124 had COPD (defined by an FEV1/FVC below the lower limit of normal). Patients with COPD were older (57 vs. 48 years), more often male (55.6% vs. 34.2%), had more smoking history (current or ex-smoker: 78.9% vs. 34.2%), and were more breathless (mMRC 4: 11.4% vs. 0%) than patients with ZZ-AATD (Table 1). In total, 111 patients had the ZZ-AATD pheno/genotype and COPD; this group was used in subsequent multivariable modelling analyses (Appendix A).

### 3.2. Biomarkers

All seven biomarkers were measured in the study cohort with no missing values. The intra-assay coefficients of variability (CV) were less than 15% across all assays except IL8 (20.1%). The inter-assay CVs for CCL18, CC16, CRP, SP-D, IL6, TNF-alpha, and IL8 were 1.2%, 6.3%, 1.8%, 3.5%, 11.5%, 10.7%, and 16.1%, respectively. The median time between biomarker measurement and baseline was 0 days (interquartile range (IQR) 2.25). There was a modest correlation (r = 0.65) between IL8 and TNF-alpha but otherwise minimal correlations between biomarkers (Appendix A). CCL18 and CRP were both significantly higher in patients with COPD compared to those without, as indicated by an unadjusted analysis (Figure 1 and Appendix A). 

### 3.3. Predicting Cross-Sectional Lung Function

No biomarkers were significantly associated with FEV1 or TL_CO_ absolute values in a multivariable model that was also adjusted for age, sex, and smoking status (never, previous, current) (Table 2). Smoking status was associated with FEV1, males had higher TL_CO_ and increasing age was associated with lower FEV1 and TL_CO_ in the same models. There was no significant difference between models that used biomarkers and those that did not for predicting absolute FEV1 (R^2^ = 0.09 vs. 0.08; *p* = 0.351) and TL_CO_ (R^2^ = 0.24 vs. 0.22; *p* = 0.241) values (Table 2 and Appendix A). Repeating this modelling with the FEV1 or TLCO percentages predicted (rather than absolute values) produced similar results and biomarkers were not significantly predictive (Appendix A).

### 3.4. Predicting Changes in Lung Function

Patients with ZZ-AATD and COPD had a median of five (IQR four) FEV1 tests and four (IQR three) TL_CO_ tests available for calculating lung function changes (Appendix A). The FEV1 (mLs/year) and TL_CO_ (mmol/min/kPa/year) absolute changes were calculated for each individual and used in multivariable modules to predict lung function changes (Appendix A).

No biomarkers were significantly associated with FEV1 absolute change in a multivariable model that was also adjusted for age, sex, and smoking status (never, previous, current) (Table 3). Smoking status and age were associated with FEV1 absolute change in this model. CC16 was associated with TL_CO_ absolute change (beta = −0.004, *p* = 0.018), with a higher CC16 associated with more TL_CO_ decline. Age was also associated with TL_CO_ absolute change in this model. The model that used biomarkers to predict TL_CO_ was significantly better than the model that did not use biomarkers (R^2^ 0.13 vs. 0.02; *p* = 0.028) (Table 3 and Appendix A). There was no significant difference between models that used biomarkers and those that did not for predicting FEV1 absolute change (*p* = 0.483). Repeating the modelling with the FEV1 or TL_CO_ percentages predicted (rather than absolute values) produced similar results, whereby the CC16 association was confirmed, and the inclusion of biomarkers had better predictive value for TL_CO_ but did not improve FEV1 prediction (Appendix A).

## 4. Discussion

AATD is an inherited disorder that causes a predisposition to emphysema-predominant COPD [2]. Identifying AATD patients at risk of accelerated lung function decline is important to allow for medical optimisation and potential selection for disease-modifying therapies such as augmentation therapy [41]. As traditional risk factors are limited in their prediction, there is a need to identify additional prognostic factors. CT density has been shown to be a robust measurement of emphysema but is not suitable pragmatically for frequent testing [13]. Similarly, exacerbation frequency has been suggested as a predictor of disease progression but is somewhat subjective and is restricted by presentation heterogeneity and reliance on self-reporting [42,43]. The use of serum biomarkers remains an attractive option as they are easily obtainable and repeatable [14]. This is particularly important with high-cost or limited-availability therapies, such as augmentation. In this study of AATD patients, we used a panel of seven serum biomarkers measurements to predict baseline and longitudinal function decline as a surrogate of disease severity.

The main finding in this study was that patients with higher serum CC16 levels had a significantly greater subsequent decline in TLco. Noteworthily, there was no significant association observed between serum CC16 concentration and FEV1, neither at baseline nor for changes over time. Whereas TLco is a measure of diffusion capacity that provides information about lung parenchymal damage, FEV1 reflects airway obstruction [44]. Therefore, our results suggest that raised serum CC16 levels in AATD are predictive of greater lung parenchymal damage due to emphysema but is not predictive of airway obstruction due to chronic bronchitis, small airway disease, or other pathology in this cohort.

CC16 is a secreted protein and is largely derived from the club cells of the bronchioles. Conflicting evidence exists regarding the role of CC16 in chronic lung disease, with evidence suggesting that the relationship between serum CC16 levels and lung function may differ between inflammatory-driven versus epithelial-driven lung diseases [45]. For inflammatory lung diseases such as usual COPD and asthma, the current consensus is that the anti-inflammatory properties of CC16 may have a protective role [46,47]. In the ECLIPSE cohort study of usual COPD, serum CC16 levels were significantly lower in COPD patients compared to controls. Levels were also lower in COPD subjects receiving long-acting β_2_ agonists. Interestingly, there was no correlation between serum CC16 levels and emphysema, although TL_CO_ was not assessed [48]. Another usual COPD study found that lower serum CC16 was associated with accelerated FEV1 decline over a nine-year period but did not investigate associations with emphysema or TL_CO_ [16]. Therefore, although the evidence suggests that CC16 may have a protective role for airway inflammation observed in COPD, there is a lack of evidence regarding the role of CC16 in emphysema. In epithelial-driven lung diseases, it is thought that CC16 may have a pathogenic role. In a study of idiopathic pulmonary fibrosis (IPF), CC16 concentrations in both serum and bronchoalveolar lavage were significantly higher in IPF patients compared with controls [49]. Similar trends have been observed in combined pulmonary fibrosis and emphysema, systemic sclerosis, and sarcoidosis, thus further supporting the evidence that raised serum CC16 levels are associated with lung parenchymal damage [50,51,52]. Despite several papers describing associations of CC16 with chronic lung disease, the underlying biological mechanism is not understood, and it remains uncertain if CC16 plays a protective or pathogenic role. Such a contrasting evidence base, coupled with the findings in this study, highlights the importance and need for further research in this area. 

In our study of AATD, the association between raised serum CC16 and greater decline in TL_CO_ was an interesting finding, given that this contrasts with previous trends observed in usual COPD. This raises interesting questions about how the differing lung pathophysiology underlying AATD COPD versus usual COPD may be affecting systemic biomarkers. To our knowledge, this is the first study investigating the role of CC16 as a biomarker in AATD patients. Another surprising finding in our study was the lack of increased serum IL6 and IL8 in those with COPD versus those without, given the abundant evidence in non-AATD COPD [22,29,30,31,32,36]. There may be several explanations for this finding, owing to the different underlying pathophysiology of AATD and non-AATD COPD. Despite not having obstructive lung disease, there may still be an element of underlying inflammation (IL6 and IL8) as a result of the AATD. Another possibility is that all participants were in a stable state of the disease at the time of serum collection and perhaps such cytokines may be more detectable at times of recent or current exacerbation [24].

Our findings highlight the importance of having dedicated research studies to investigate this unique cohort, as trends observed in usual COPD may not be translated to AATD COPD. This may also explain why the other biomarkers in our study, all of which have been shown to be associated with lung function in usual COPD, were not associated with lung function in this AATD COPD cohort. Our study did show that subjects with AATD COPD had significantly higher serum CCL18 and CRP levels than AATD patients without COPD. This finding was unsurprising and likely reflects the chronic inflammation observed in COPD, and lack of AAT, failing to dampen inflammation initiated by protease imbalance. It does, however, add to the assumption that lung pathophysiology can be assessed with blood biomarkers. Alternative sampling modalities could be used in future work, such as the measurement of biomarkers in sputum or through immunofluorescent labelling of formalin-fixed paraffin-embedded lung tissue. In fact, multiple studies have demonstrated significant results in the sputum of patients with COPD, which may be considered more representative of the airways [53,54,55]. This method is limited by the fact that only certain patients produce spontaneous sputum, particularly in AATD COPD, thereby limiting the cohort. For similar reasons, a “healthy” control that produce sputum in the absence of lung disease, cannot be sought. Therefore, the clinical applicability of the sampling modality must be an important consideration. 

Our cohort was representative of a moderate–severe COPD population, with almost 90% of patients being in the GOLD II status category or above. Interestingly, biomarkers were not associated with COPD severity (as represented by FEV1) in our study (Table 2 and Table 3). One possible explanation for this is that those with an already-low FEV1 may have a slower rate of decline, given they have less lung volume to lose. That also infers that rapid decliners are unlikely to have severe COPD at baseline. Similarly, biomarkers may be less associated with FEV1 in severe COPD if destruction of the airways has already occurred and has since plateaued [24]. 

Biomarkers are commonly altered in the blood of those with prolonged cigarette exposure versus those without, particularly in relation to inflammatory cytokines [56]. Almost three-quarters of our investigated cohort with AATD COPD had a history of cigarette smoke exposure, with a further 4% being current smokers at the time of sampling (Table 1). We, therefore, mitigated the effect of smoking on blood biomarkers by including smoking status as a covariate in our multivariate prediction models. Whilst previous smokers had a significantly reduced baseline FEV1 and change in FEV1 over time (Table 2 and Table 3, respectively) the main findings we report above were independent of smoking status. No association was observed between current smokers and baseline/changes in FEV1, which may represent a potential false negative association attributable to a small group size. Another consideration and nuance of our smoking status classification (never smokers, previous smokers and current smokers) was the lack of representation of smoking duration, assumption of cigarette-smoke exposure (as opposed to other inhaled smoke), or variable depth of inhalation between subjects—all of which may influence disease progression (and hence, lung function parameters) [57,58]. 

A study strength is that we used multiple rather than single biomarkers in a rare disease cohort combined with traditional risk factors to predict disease severity in the form of cross-sectional and longitudinal lung function. Limitations of our study include the retrospective design, variable follow-up durations, and the recruitment of patients from a single AATD registry. The number of patients enrolled in this study was largely constrained by disease rarity, thus resulting in insufficient patient numbers for a validation cohort. We, therefore, recommend further confirmation of our CC16 findings. Incorporating longitudinal biomarker measurements may also increase their predictive value. Whilst our biomarker panel incorporated a selection of the most robustly validated serum biomarkers for usual COPD, there may be alternative blood biomarkers that were not included in our study with better predictive accuracy. Such biomarkers, including fibrinogen, soluble receptor for advanced glycation end products (sRAGE), and mixed metalloproteins (MMPs) have all demonstrated potential in usual COPD but were not included in our study [59,60,61]. To avoid the selection bias associated with candidate biomarker selection, proteomics is a promising alternative approach that warrants further research. Proteomics has been successfully employed in both AATD COPD and usual COPD to predict biomarkers associated with emphysema, the GOLD stages, and FEV [62,63]. A recent multi-centre, proteomics study in AATD COPD revealed 98 proteins that were associated with diffusing capacity of the lungs for carbon monoxide in PiZZ AATD, thereby potentiating suitable candidates for further exploration [63]. 

In conclusion, this study of AATD patients assessed the ability of a panel of seven serum biomarkers to predict baseline and longitudinal function decline. The main study finding was that patients with higher serum CC16 levels had a significantly greater subsequent decline in TLco. This contrasts with previous trends observed in usual COPD and highlights the importance of dedicated research studies for this unique cohort. Further research is required to validate these findings and we also recommend proteomics as a valuable tool going forward to identify additional biomarkers for AATD. 

## Figures and Tables

**Figure 1 biomedicines-11-02001-f001:**
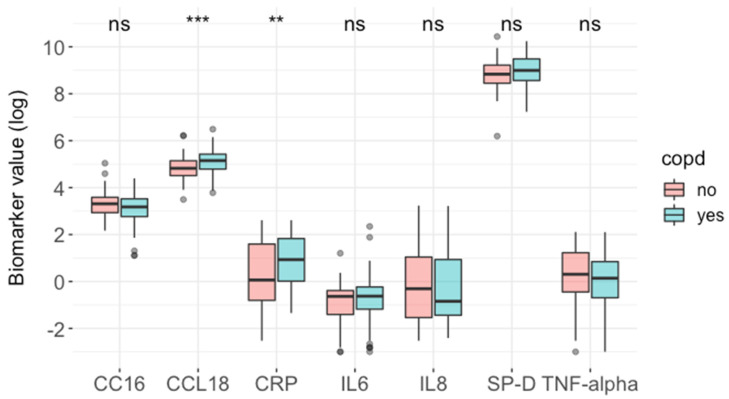
Blood biomarkers in AATD patients with and without COPD. Blood biomarkers were log-transformed (to aid visualisation) and compared between AATD patients with (*n* = 124) and without (*n* = 76) COPD using the Mann–Whitney U test. *p*-values: *** < 0.001, ** <0.01; ns, non-significant. CC16, club cell secretory protein-16 (ng/mL); CCL18, chemokine ligand 18 (ng/mL); CRP, C-reactive protein (mg/L); IL6, interleukin 6 (pg/mL); IL8, interleukin 8 (pg/mL); SP-D, surfactant protein D (pg/mL); TNF-alpha, tumour necrosis factor alpha (pg/mL).

**Table 1 biomedicines-11-02001-t001:** Characteristics of patients with and without COPD. Values are presented as median [IQR] or n (%). COPD, chronic obstructive pulmonary disease; FEV1, forced expiratory volume in 1 s; FVC, forced vital capacity; mMRC, modified Medical Research Council dyspnoea scale; TL_CO_, transfer factor of the lung for carbon monoxide; NA, not applicable.

	No COPD (*n* = 76)	COPD (*n* = 124)
**Age, years**	48 [40–59]	57 [48–64]
**Sex, male**	26 (34.2)	69 (55.6)
**Follow-up, years**	4.18 [2.80–7.53]	5.47 [2.98–8.63]
**Smoking status** Current Previous Never	3 (3.9) 23 (30.3) 50 (65.8)	5 (4.1) 92 (74.8) 26 (21.1)
**Genotype** ZZ SZ Other	56 (73.7) 18 (23.7) 2 (2.6)	111 (89.5) 6 (4.8) 7 (5.6)
**GOLD COPD stage** I II III IV	NA NA NA NA	15 (12.1) 48 (38.7) 41 (33.1) 20 (16.1)
**FEV1, litres**	3.36 [2.80–3.94]	1.60 [1.15–2.06]
**FEV1 % predicted**	104 [94–112]	54 [36–69]
**FVC, litres**	4.19 [3.45–5.05]	3.75 [3.04–4.73]
**FVC % predicted**	103 [94–112]	97 [80–111]
**FEV1/FVC**	0.76 [0.75–0.84]	0.44 [0.33–0.53]
**TL_CO_, mmol/min/kPa**	7.68 [6.69–9.81]	4.86 [3.79–6.46]
**TL_CO_ % predicted**	100 [90–114]	64 [48–78]
**mMRC** 0 1 2 3 4	32 (69.6) 10 (21.7) 4 (8.7) 0 (0) 0 (0)	9 (12.9) 14 (20.0) 20 (28.6) 19 (27.1) 8 (11.4)

**Table 2 biomedicines-11-02001-t002:** Multivariable model to predict absolute FEV1 or TL_CO_ values at baseline in patients with ZZ-AATD and COPD. FEV1 or TL_CO_ absolute values were modelled using the equation: FEV1 or TL_CO_ ~ Age + Sex + Smoking + Biomarkers (CC16 + CCL18 + CRP + IL6 + IL8 + TNF-alpha + SP-D). The beta coefficients (Beta1 and Beta2), standard error (SE1 and SE2), *p*-values (P1 and P2), adjusted R^2^ (R^2^), and number included in each model (n) are presented in the table. Beta coefficients represent the direction and magnitude of effects and *p*-values represent whether the variable is significantly associated with the outcome variable (e.g., a beta coefficient of −0.009 for age means that for every 1 unit increased in age, FEV1 decreases by 0.009 units). Standard error is a measure of the statistical accuracy of the estimate (beta), with a higher value denoting a less accurate estimate. For categorical variables, the beta coefficients displayed are with respect to a reference: for sex, reference is female and for smoking status, reference is never-smokers. This explanation applies to all multivariate models in this study.

	FEV1 (*n* = 110) R^2^ = 0.09		TL_CO_ (*n* = 98) R^2^ = 0.24	
	Beta1 (SE1)	P1	Beta2 (SE2)	P2
**Age**	−0.009 (0.007)	0.243	−0.082 (0.019)	<0.001
**Male**	0.263 (0.146)	0.075	1.197 (0.408)	0.004
**Previous smoker**	−0.484 (0.174)	0.007	−0.124 (0.465)	0.791
**Current smoker**	−0.765 (0.535)	0.156	−0.744 (1.381)	0.592
**CC16**	0.004 (0.006)	0.476	0.016 (0.015)	0.302
**CCL18**	−0.001 (0.001)	0.207	−0.002 (0.002)	0.301
**CRP**	−0.033 (0.021)	0.125	−0.052 (0.056)	0.347
**IL6**	−0.038 (0.069)	0.583	−0.205 (0.182)	0.262
**IL8**	−0.030 (0.026)	0.246	0.045 (0.067)	0.505
**TNF-alpha**	0.089 (0.071)	0.216	0.172 (0.197)	0.387
**SP-D**	<0.001 (<0.001)	0.446	<0.001 (<0.001)	0.625

**Table 3 biomedicines-11-02001-t003:** Multivariable model to predict absolute FEV1 or TL_CO_ changes over time in patients with ZZ-AATD and COPD. FEV1 (mLs/year) or TL_CO_ (mmol/min/kPa/year) absolute changes were modelled using the equation: FEV1 or TL_CO_ change ~ Age + Sex + Smoking + Biomarkers (CC16 + CCL18 + CRP + IL6 + IL8 + TNF-alpha + SP-D). The beta coefficients (Beta1 and Beta2), standard error (SE1 and SE2), *p*-values (P1 and P2), adjusted R^2^ (R^2^), and number included in each model (n) are presented in the table.

	FEV1 change (*n* = 94) R^2^ = 0.14		TL_CO_ change (*n* = 84) R^2^ = 0.18	
	Beta1 (SE1)	P1	Beta2 (SE2)	P2
**Age**	3.624 (0.827)	<0.001	0.008 (0.002)	0.002
**Male**	3.159 (15.765)	0.842	−0.041 (0.049)	0.407
**Previous smoker**	45.004 (19.174)	0.021	0.026 (0.057)	0.644
**Current smoker**	22.801 (72.797)	0.755	−0.016 (0.212)	0.938
**CC16**	−0.639 (0.576)	0.270	−0.004 (0.002)	0.018
**CCL18**	0.082 (0.086)	0.347	<0.001 (<0.001)	0.414
**CRP**	0.003 (2.222)	0.999	−0.006 (0.007)	0.401
**IL6**	2.531 (6.858)	0.713	−0.005 (0.020)	0.811
**IL8**	−2.361 (2.543)	0.356	0.012 (0.008)	0.127
**TNF-alpha**	8.107 (7.730)	0.297	−0.005 (0.023)	0.815
**SP-D**	0.001 (0.001)	0.604	<0.001 (<0.001)	0.367

## Data Availability

Data is contained within the article or Appendix A.

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
