# Peer review of "Predicting Lung Function Using Biomarkers in Alpha-1 Antitrypsin Deficiency"

_biomedicines, 2023, doi:10.3390/biomedicines11072001_

Round 1
Reviewer 1 Report
In a clinical study, the authors investigate serum protein markers regarding their predictive power to assess severity of COPD and lung damage in patients with alpha-1 antitrypsin deficiency. They determined change in FEV1 and diffusing capacity of the lungs for carbon monoxide in a longitudinal study as a gold standard method to predict and establish severity of COPD or lung parenchymal damage and estimate the decline in lung function during the study. They use a multivariable model and statistical analysis to draw correlations between serum markers and results from lung function tests to present serum marker usefulness in predicting the severity of lung damage. They found that CC16 protein could be a valuable tool for assessing lung damage in AATD with COPD, although a multi-center study would be necessary to confirm this finding. The methods and statistical analysis used in the study are appropriate. The results support the conclusions.
Suggestions:
1) Under the section on Biomarkers, the authors state “There was modest correlation (r=0.65) between IL6 and TNF-alpha but otherwise minimal correlation between biomarkers (Figure S2).” I believe this correlation presented between IL8 and TNF-alpha instead of IL6 based on Figure S2. Please correct it.
2) Under the Discussion section, please add a reference to the statement, “Similarly, exacerbation frequency has been suggested as a predictor of disease progression but is somewhat subjective and is restricted by presentation heterogeneity and reliance on self-reporting.”
3) Also, in the Discussion, please correct the sentence, “To our knowledge, this is the first study investigating the role of CC19 as a biomarker in AATD patients.” I believe CC16 should be used instead of CC19.
Author Response
Dear Reviewer,
We thank you for taking the time to review our manuscript and appreciate the comments you have made. Please see below the revisions we have made, in relation to your comments:
“1) Under the section on Biomarkers, the authors state “There was modest correlation (r=0.65) between IL6 and TNF-alpha but otherwise minimal correlation between biomarkers (Figure S2).” I believe this correlation presented between IL8 and TNF-alpha instead of IL6 based on Figure S2. Please correct it.” Thank you for highlighting this, we have amended accordingly.
“2) Under the Discussion section, please add a reference to the statement, “Similarly, exacerbation frequency has been suggested as a predictor of disease progression but is somewhat subjective and is restricted by presentation heterogeneity and reliance on self- reporting.”” We have added a reference to support this statement.
“3) Also, in the Discussion, please correct the sentence, “To our knowledge, this is the first study investigating the role of CC19 as a biomarker in AATD patients.” I believe CC16 should be used instead of CC19.” Thank you for highlighting this, we have amended accordingly.
Kindest regards,
Daniella Spittle
Reviewer 2 Report
This manuscript described various potential biomarkers for the AATD-associated COPD development. Authors also trying to connect declined lung function and upregulated biomarkers in patient cohort. The author have detected several inflammatory biomarkers, such as IL-6, IL-8, CC-16, CRP, TNF-a, and SP-D. Although there is no correlation between cytokine levels and lung function declining, CC-16 was associated with transfer factor for carbon monoxide. Some minor suggestions for authors to consider to improve the manuscript, especially focused in discussion.
The authors discussed majorly cc-16 with lung disease, however, other inflammatory cytokines were common upregulated during lung injury, such as IL-6 and IL-8. Also, cigarette smoking should be considered as it is the major cause of emphysema/COPD. Correlation of different cytokines from human sample cohort (smoker vs health, PMID: 31494573 ) can be added into the discussion panel to increase the translational value of how cytokines are potentially serving as biomarkers for COPD development.
Author Response
Dear Reviewer,
We thank you for taking the time to review our manuscript and appreciate the comments you have made. Please see below the revisions we have made, in relation to your comments:
“The authors discussed majorly cc-16 with lung disease, however, other inflammatory cytokines were common upregulated during lung injury, such as IL-6 and IL-8.” We agree with the reviewer that our lack of this finding in our cohort was surprising, given the strong evidence of association in non-AATD COPD. We have addressed this in the discussion with suggestions as to why this may have been the case.
“Also, cigarette smoking should be considered as it is the major cause of emphysema/COPD. Correlation of different cytokines from human sample cohort (smoker vs health, PMID: 31494573) can be added into the discussion panel to increase the translational value of how cytokines are potentially serving as biomarkers for COPD development.” We have added a paragraph to the discussion section to describe the influence that cigarette smoke exposure may have on biomarker concentrations in COPD. We have included the suggested reference (PMID: 31494573).
Kindest regards,
Daniella Spittle
Reviewer 3 Report
The report by Daniella Spittle et al. analysed 7 biomarkers in the blood of alpha-1 antitrypsin deficiency (AATD) patients (n=200) with and without COPD. The manuscript is well-written and well-structured. The topic is relevant in clinical investigations. Nonetheless, the lack of experimental data and validations weakens the general interest in the study.
Major comments:
1. Introduction: the background justifying the 7 biomarkers is not sufficient. The authors should better explain how the “selection of the most robustly validated serum biomarkers for COPD” was performed.
2. A lot of references appeared outdated and should be updated. For instance: 16, 23, 24, 25, 28, 29, 30.
3. Methods: (i) is it possible to describe the severity of COPD? Is there an influence regarding biomarkers’ detection/concentration? (ii) the selection of the ELISA kits should be justified (sensitivity? Literature? Etc.)
4. Results: (i) the smoking history appears as a huge bias in the analysis. How do the authors distinguish between AATD and smoking effects? Did the authors check separately the blood biomarkers in ex vs never smokers in COPD and non-COPD groups to evaluate this aspect? (ii) regarding the lung function changes: was the period standardized (in months or years)? I would assume that the total period that was analysed including repeated tests influenced the outcome.
5. Figure 1: it would be useful to present the concentrations rather than log-transformed data since the readers may want to gather quantitative data on these biomarkers.
6. The paper would gain strength if additional experimental data were given. For example, the detection of CC16 (and the other biomarkers) could be tested in IF of FFPE lung tissues of AATD patients with/without COPD if available.
Minor comments:
1. The title is somehow misleading as the results more likely demonstrated that using biomarkers to predict lung function is difficult.
2. The third sentence of the discussion is missing punctuation.
3. The fourth reference of the discussion has no number.
4. The fourth paragraph mentioned CC19 instead of CC16.
5. The alternative additional biomarkers should be named and discussed.
6. Considering that only 2 patients were current smokers, the line integrating this aspect in the statistic models appears unnecessary.
Author Response
Dear Reviewer,
We thank you for taking the time to review our manuscript and we appreciate the comments you have made. Please see below our revisions, in relation to your comments:
“1. Introduction: the background justifying the 7 biomarkers is not sufficient. The authors should better explain how the “selection of the most robustly validated serum biomarkers for COPD” was performed.” We have added great detail to the introduction section to discuss evidence behind the seven selected biomarkers.
“2. A lot of references appeared outdated and should be updated. For instance: 16, 23, 24, 25, 28, 29, 30.”
Thank you for your suggestion. We have re-reviewed the literature and we have updated to more contemporaneous references where appropriate and have also retained important references where they are landmark studies or where they represent the best evidence available.
- Reference 16 (now reference 32): Pinto-Plata et al 2007 is a unique study showing an association between IL8/TNF-alpha and TLCO. We have retained this reference, but we have changed the text to explicitly clarify this and make it more specific.
- Reference 23 (now reference 44): Gioldassi et al 2004 has been replaced with Li et al 2023, which uses newer CC16 mRNA expression techniques.
- Reference 24 (now reference 47): Lomas et al 2008 is part of the landmark ECLIPSE study which we feel is a reputable study with a large cohort. Based on this, we have retained the reference.
- Reference 25 (now reference 16): Park et al 2013 is one of a limited number of studies investigating associations between CC16 and FEV1 decline in usual COPD. For this reason, we have decided to retain this reference.
- Reference 28 (now reference 51): Hasegawa et al 2011 is the only study investigating associations between CC16 and pulmonary fibrosis in systemic sclerosis. We have therefore decided to leave this reference as is.
-
- Reference 29 (now reference 52): Hermans et al 2001 has been replaced with Kucejko et al 2009, which is a more recent study investigating associations between CC16 and pulmonary fibrosis in sarcoidosis.
- Reference 30: Janssen et al 2003 has been removed, as the above reference (Kucejko et al 2009) is a suitable and sufficient paper.
“3. Methods: (i) is it possible to describe the severity of COPD? Is there an influence regarding biomarkers’ detection/concentration?
We have described the GOLD COPD stages and added them to table 1 for AATD-COPD patients. The multivariable models are assessing association between biomarkers and either FEV1 or TLCO, which also addresses the association of biomarkers with COPD severity. We have used FEV1 as a continuous metric rather than classified it into GOLD COPD groups for the modelling as this is a more robust statistical approach. The beta coefficients in the tables represent the magnitude and direction of effect. For example, in Table 2 a negative beta-coefficient relates to a lower FEV1 or TLCO for a change in that variable. Therefore, no biomarkers were significantly associated with FEV1 (COPD severity) or TLCO at baseline.We have clarified what the beta-coefficients relate to in the table legends. We have also amended the discussion in line with your suggestion to clarify the association between COPD severity and biomarkers.
(ii) the selection of the ELISA kits should be justified (sensitivity? Literature? Etc.)” We have added the following sentence to the methods section to justify our ELISA kit selection: “ELISA kits were selected with appropriate sensitivity, according to expected serum concentrations from the literature”.
“4. Results: (i) the smoking history appears as a huge bias in the analysis. How do the authors distinguish between AATD and smoking effects? Did the authors check separately the blood biomarkers in ex vs never smokers in COPD and non-COPD groups to evaluate this aspect? (ii) regarding the lung function changes: was the period standardized (in months or years)? I would assume that the total period that was analysed including repeated tests influenced the outcome.” The multivariable models are primarily to assess the association between biomarkers and FEV1/TLCO. They have been adjusted for other variables that could confound/bias the analysis and influence FEV1/TLCO, which includes age, sex, and smoking status. Smoking status is a categorical variable which contains never-smoked, ex-smoker and current smoker, which were included in the model. Including smoking status in the multivariable models made the analysis more robust by accounting for potential confounding effects that smoking could have on FEV1 (and biomarkers). As highlighted, some groups (i.e., current smokers) contained a smaller number of individuals, however this did not significantly influence the analysis as group size is considered in the modelling and reflected by the accuracy of the estimate (standard error). We addressed this comment by i) adjusting the manuscript to make the model interpretation clearer ii) expanding the discussion to specifically address smoking status and iii) including some categorical group sizes as limitations in the discussion. Lung function change (in FEV1/TLCO) was standardised by calculating the slope of change in patients with 3 or more lung function tests which were then used in the multivariable models. This enabled standardisation of lung function change across individuals as patients had lung function recorded at different time intervals. We have also included supplementary figures of the lung function change in FEV1 and TLCO per year for further information.
“5. Figure 1: it would be useful to present the concentrations rather than log-transformed data since the readers may want to gather quantitative data on these biomarkers.” Thank you for your suggestion. We have now included a supplementary table (S2) with the raw values of biomarker concentrations. We have presented the data in a table as a figure of the raw values was not visually interpretable due to the differing biomarker value scales. We have retained figure 1 as the log-transformed values aid visualisation and comparison between groups without the transformation process changing the associations.
“6. The paper would gain strength if additional experimental data were given. For example, the detection of CC16 (and the other biomarkers) could be tested in IF of FFPE lung tissues of AATD patients with/without COPD if available.” Whilst we appreciate the strength that additional experimental data would give our article, this is beyond the capabilities of our research cohort. We have recognised this in the discussion section by adding a paragraph mentioning alternative sampling modalities, including IF of FFPE lung tissue.
“1. The title is somehow misleading as the results more likely demonstrated that using biomarkers to predict lung function is difficult.” The authors preference would be for the title to remain unchanged. We believe the title does not assert that it is possible to predict lung function with biomarkers, but instead highlights the key, succinct aim of our study.
“2. The third sentence of the discussion is missing punctuation.” Thank you for highlighting this, we have amended accordingly.
“3. The fourth reference of the discussion has no number.” We have corrected this mistake and have added the appropriate references.
“4. The fourth paragraph mentioned CC19 instead of CC16.” Thank you for highlighting this, we have amended accordingly.
“5. The alternative additional biomarkers should be named and discussed.” We have added detail to this point in the penultimate paragraph of the discussion. This has included naming alternative biomarkers and expanding on the discussion of proteomics as a way of identifying candidate biomarkers in future work.
“6. Considering that only 2 patients were current smokers, the line integrating this aspect in the statistic models appears unnecessary.” We have addressed this point as detailed above – we amended the manuscript to justify the rationale for including smoking status and extended the discussion to address the association with smoking status and limitations of small group sizes.
Kindest regards,
Daniella Spittle
Round 2
Reviewer 3 Report
The authors addressed all the comments satisfactorily. I have no additional remark.